# Interspecies interaction reduces selection for antibiotic resistance in *Escherichia coli*

Ramith R. Nair [1✉] & Dan I. Andersson [1]

Evolution of microbial traits depends on the interaction of a species with its environment as well as with other coinhabiting species. However, our understanding of the evolution of specific microbial traits, such as antibiotic resistance in complex environments is limited. Here, we determine the role of interspecies interactions on the dynamics of nitrofurantoin (NIT) resistance selection among *Escherichia coli*. We created a synthetic two-species community comprised of two variants of *E. coli* (NIT susceptible and resistant) and *Bacillus subtilis* in minimal media with glucose as the sole carbon source. We show that the presence of *B. subtilis* significantly slows down the selection for the resistant *E. coli* mutant when NIT is present and that this slowdown is not due to competition for resources. Instead, the dampening of NIT resistance enrichment is largely mediated by extracellular compounds produced by *B. subtilis* with the peptide YydF playing a significant role. Our results not only demonstrate the impact of interspecies interactions on the evolution of microbial traits but also show the importance of using synthetic microbial systems in unravelling relevant interactions and mechanisms affecting the evolution of antibiotic resistance. This finding implies that interspecies interactions should be considered to better understand and predict resistance evolution in the clinic as well as in nature.

[1] Department of Medical Biochemistry and Microbiology, Uppsala University, Uppsala SE-75123, Sweden. ✉email: ramith_nair@hotmail.com

Antibiotic resistance is a medically important bacterial trait that can arise in a population as a result of de novo mutations or by horizontal gene transfer. Resistant strains typically rapidly outcompete susceptible strains at concentrations above the minimal inhibitory concentration (MIC) leading to dominance or fixation of the trait[1,2]. However, the emergence and subsequent enrichment of a resistant strain towards dominance or fixation have also been observed at antibiotic concentrations far below the MIC, termed sub-MIC[1,3–8]. Bacteria can be exposed to sub-MIC levels of antibiotics in clinical settings during therapeutic use[9,10] as well in the environment (e.g., water, soils, and food) due to anthropogenic pollution[9,11,12]. Bacterial growth at sub-MIC not only broadens the range of probable mutations leading to antibiotic resistance but the weaker selective pressure by the antibiotic also affects the rate of resistance enrichment[3,7].

Emergence and enrichment of antibiotic resistance at sub-MIC have been studied quite extensively over the last few decades[1,5,8,13–20]. However, a majority of these studies have focused on single-species systems, overlooking the fact that bacteria often exist as part of a consortium comprised of many microbial species. The biotic and abiotic interactions a bacterial species is exposed to could influence resistance selection through a range of different mechanisms. For example, competitive interactions between different microbes will reduce the availability of shared resources, which should negatively affect population density, reducing the number of possible doublings of the resistant strain thereby influencing its enrichment in the presence of antibiotics. Alternatively, microbial species can also partake in positive interactions through cross-feeding interactions, where the two species are mutually dependent on each other for essential metabolites[21], which will increase the population density and the number of generations that could affect the emergence and enrichment of antibiotic resistance. Also, the presence of another species in the ecosystem might result in sequestration of antibiotics present in the environment, which will reduce the level of available antibiotics affecting bacterial growth, thereby slowing the rate of selection. Finally, some bacterial species are also known to deactivate antibiotics (e.g., degradation of beta-lactams by beta-lactamases[22–24]), which is also expected to reduce the rate of enrichment of the resistant strain.

In the context of antibiotic resistance, the presence of a consortium of species from pig feces has been shown to offset the relative fitness advantage of kanamycin and rifampicin-resistant strains in the presence of a range of antibiotic concentrations[20]. Also, the presence of Acetobacter species was shown to improve the survival of susceptible Lactobacillus cells in the presence of rifampin[25]. Other examples have established the effect of interspecies interactions on the modulation of relative fitness of resistant strains in the presence of antibiotics (reviewed in ref. [26]), but in a majority of cases, the mechanisms underlying these effects remain undefined, partly due to the difficulties in identifying and manipulating interactions in complex natural communities[20,27,28].

Here, we designed a defined and genetically amenable synthetic community to study the impact of species interactions on antibiotic resistance dynamics. To this end, we used Escherichia coli MG1655 and Bacillus subtilis 168 to study the selection for nitrofurantoin (NIT) resistant E. coli at sub-MIC NIT levels. B. subtilis is a Gram-positive bacterium that has found a wide variety of uses as a cell factory[29], animal feed additive[30], probiotic[31], as well as a probable antigen delivery vector to the mucosal layer in the gut[32]. The use of the bacterium in food as well as a vector to the gut might bring it into direct contact with the gut microflora, where E. coli is generally present. Furthermore, B. subtilis can in immunocompromised people cause several different types of infections[33,34], implying that co-infections

of E. coli and B. subtilis could occur. NIT is often used for the treatment of E. coli urinary tract infections[35,36]. NIT is a pro-drug that is activated inside the E. coli cells through nitroreductases, predominantly NfsA and NfsB, forming a free radical[37] that is thought to damage DNA[38,39] and ribosomes[40]. Loss of genes encoding these enzymes renders E. coli resistant to NIT[41].

In this study, we followed the enrichment of NIT-resistant (NIT$^R$) E. coli in a mixed population of susceptible and resistant E. coli in the presence of another species, B. subtilis. We show that the rate of enrichment of the NIT$^R$ E. coli cells at sub-MIC is impeded by the presence of B. subtilis. Further, our results show that the impediment is due to interference competition between the two species, largely mediated by extracellular molecules excreted by B. subtilis with the peptide, YydF, playing a key role.

## Results

**Species interactions slow nitrofurantoin resistance enrichment at sub-MIC.** Using the broth microdilution method, we determined the MIC of NIT for both E. coli and B. subtilis to be 4 mg/L (Supplementary Fig. 1) in minimal media with glucose as the sole carbon source. Next, we elucidated the role of interspecies interactions on the selection of NIT resistance at sub-MIC (1/4, 1/6, and 1/8 of MIC of the susceptible strain) by competing susceptible and resistant (ΔnfsAB) fluorescent strains of E. coli. Starting from an initial frequency of ~0.01 (1%), the resistant cells enriched to 0.5–0.75 in frequency within three cycles of 24-h competitions in the absence of interactions with B. subtilis, compared to 0.05–0.15 in its presence (Fig. 1a), demonstrating a significant interaction induced dampening of NIT$^R$ enrichment at sub-MIC (Fig. 1b). B. subtilis and E. coli both coexisted stably during the course of the experiment (Supplementary Fig. 2).

**NIT$^R$ enrichment is unaffected by resource competition.** Competitive interaction between two species can take the form of scramble competition (where there is no direct interaction between the two species) or interference competition (where one species directly inhibits the other species)[42]. E. coli and B. subtilis are both capable of consuming glucose as a carbon source, while lactose and salicin can only be consumed by E. coli and B. subtilis, respectively. To examine the kind of interaction dominating the evolutionary dynamics of NIT resistance in our system, we performed the competitions in the presence of (i) glucose, (ii) lactose and salicin, or (iii) all three carbon sources. If the observed change in dynamics of resistance enrichment is largely due to scramble competition, we expect that to be mitigated when species are not involved in a competition for resources (when glucose is not provided as a carbon source) as compared to regimes when glucose was present during part or the entire growth cycle. In the presence of B. subtilis, the rate of resistance enrichment was similar among the three regimes (Fig. 2a) with no significant difference in mean log-transformed relative fitness after three rounds of competitions (Fig. 2b, ANOVA: $F_{2,45} = 3.215$, $P = 0.0495$; Tukey's HSD test: LS vs G, $P = 0.12$; GLS vs G, $P = 0.94$; GLS vs LS, $P = 0.06$) suggesting a predominant role of interference competition in driving the observed slowdown of NIT resistance enrichment (initial and final population frequencies were used to determine relative fitness estimates over 3 days as per ref. [43]).

**Heat-sensitive compounds in the B. subtilis culture supernatant are largely responsible for dampening NIT$^R$ enrichment.** Among bacterial species, interference competition between species is frequently mediated by extracellular compounds[44]. We examined if that was the case here by elucidating the dynamics of NIT$^R$ enrichment as well as its relative fitness in the presence of

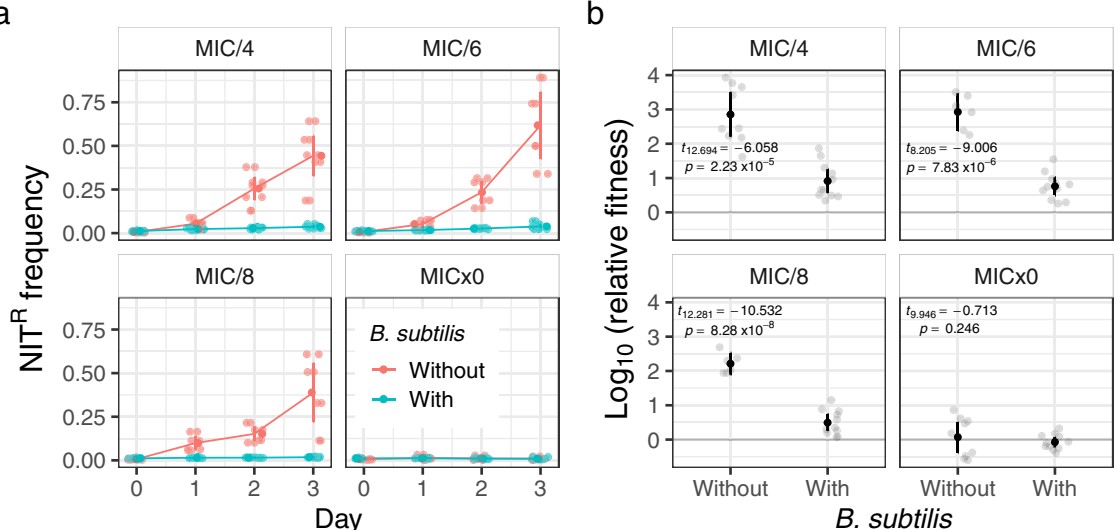

**Fig. 1 *B. subtilis* dampens NIT resistance enrichment in *E. coli*. a** Lines depict increases in frequencies of NIT-resistant (NIT$^R$) cells over 3 days in the presence of different sub-MIC levels of NIT. **b** Depiction of log-transformed relative fitness of NIT$^R$ cells in the presence and absence of *B. subtilis* after 3 days of competition with NIT$^S$ cells. Values on each plot represent those after Welch's one-sided *t* test between log-transformed relative fitness estimates of resistant strain in the presence and absence of *B. subtilis* for each antibiotic concentration. The log-transformed relative fitness value of zero (gray horizontal line) represents no fitness advantage to either NIT$^R$ or NIT$^S$ strains, while positive values indicate an advantage to NIT$^R$ strains. Lighter points depict individual replicates, darker points represent means and error bars represent 95% confidence intervals (*t*-distribution, *n* = 12 for both panels).

cell-free supernatant from a fully grown *B. subtilis* culture. We found that the relative fitness estimate of the resistant strain after three rounds of 24-h competitions to be significantly reduced in the presence of the *B. subtilis* supernatant (Fig. 3b, ANOVA: $F_{2,33} = 128.8$, $P = 2.57 \times 10^{-16}$; Tukey's HSD test: PBS vs Filtered, $P = 5.91 \times 10^{-14}$), which reduces the rate of NIT$^R$ enrichment (Fig. 3a). In addition, we found that the observed reduction is significantly offset with heat-treated supernatant (Fig. 3, Tukey's HSD test: PBS vs Filtered+Heated, $P = 2.09 \times 10^{-12}$; Filtered+ Heated vs Filtered, $P = 0.0005$). Thus, *B. subtilis* makes heat-sensitive compounds that affect the enrichment of NIT resistance in the presence of sub-MIC of NIT. Further, we also found that reducing the number of possible *E. coli* generations through dilution of growth media in half with phosphate-buffered saline did not affect the enrichment or relative fitness of NIT$^R$ strain (Supplementary Fig. 3, Welch two-sample *t* test: $t_{18.081} = 1.2833$, $P = 0.216$) further pointing to the possibility of interference by *B. subtilis* population in dampening NIT$^R$ enrichment.

**B. subtilis extracellular peptide YydF dampens *E. coli* NIT$^R$ enrichment.** Apart from significantly affecting NIT$^R$ enrichment, *B. subtilis* culture supernatant also inhibited *E. coli* growth in minimal media (Fig. 4), so we set out to identify the component in the supernatant that might result in this inhibition with the assumption that the same component might be responsible for the change in *E. coli* NIT resistance enrichment. The approach we used to identify the component in the supernatant that results in this growth inhibition is summarized in Fig. 5. Briefly, we split the supernatant into two fractions—fraction 1 (with components of MWs >20 KDa) and fraction 2 (with components between MWs 3–20 KDa, see "Methods"). We found that fraction 2 results in significant inhibition of *E. coli* growth (Supplementary Fig. 4b). Further, we found the growth inhibition of *E. coli* to be highly variable between replicates (Supplementary Fig. 5a, b), which allowed us to identify three supernatants (among >20 tested) that inhibited *E. coli* growth to different extents (Supplementary Fig. 5c). Thus, we assumed that the degree of inhibition was related to the amount of the unknown protein components.

Comparison of protein components in these supernatants by mass spectrometry revealed YydF and YorD to be proteins of interest (see "Methods" for details of the comparison).

To determine if the production of these proteins by *B. subtilis* did dampen NIT$^R$ enrichment, we competed resistant and susceptible *E. coli* populations in the presence of *B. subtilis* strains knocked out for the genes encoding these proteins. The relative fitness of NIT$^R$ *E. coli* was found to be significantly higher in the presence of *B. subtilis* $\Delta yydF$ compared to that in the presence of wild-type strain (Fig. 6b, ANOVA: $F_{3,24} = 10.9$, $P = 0.0001$; Tukey's HSD test: $\Delta yydF$ vs WT, $P = 0.017$), exhibiting significant role of the protein in the dampening of NIT resistance enrichment. However, though the average relative fitness of the resistant *E. coli* population was found to be higher in the presence of *B. subtilis* $\Delta yorD$ strain (compared to that in the presence of wild-type strain), the difference was found to be statistically insignificant (Fig. 6b, Tukey's HSD test: $\Delta yorD$ vs WT, $P = 0.119$), indicating a possible minor role of the protein in dampening NIT$^R$ enrichment (Fig. 6).

**Effects of YydF pre-pro-peptide and YydF* epipeptide on *E. coli* growth.** Previous studies have shown the peptide YydF to be a 49 amino acid-long pre-pro-peptide that undergoes multiple modifications before being secreted by *B. subtilis* cells as a 17-mer epipeptide YydF*, which is capable of killing *B. subtilis* by dissipating its membrane potential[45]. Since the molecular weight of YydF*, at 2.1 KDa, is lower than the components in fraction 2, we first sought to determine if the 49-mer pre-pro-YydF was capable of affecting *E. coli* growth in minimal media thereby resulting in previously observed dampening of NIT$^R$ enrichment. In the presence of NIT, log-transformed relative OD$_{600}$ (ratio of *E. coli* OD$_{600}$ in the presence over that in the absence of pre-pro-YydF after 24 h of growth) was significantly lower in the NIT$^R$ strain as compared to that of NIT susceptible (NIT$^S$) strain at three of the four protein concentrations we tested (Supplementary Fig. 6, Welch's one-sided *t* test: $t_{9.87} = -2.199$, $P = 0.026$ (8.75 nM); $t_{13.158} = -4.451$, $P = 0.0003$ (17.5 nM); $t_{8.633} = -5.245$, $P = 0.0003$ (35 nM) and $t_{8.689} = -3.252$, $P = 0.005$

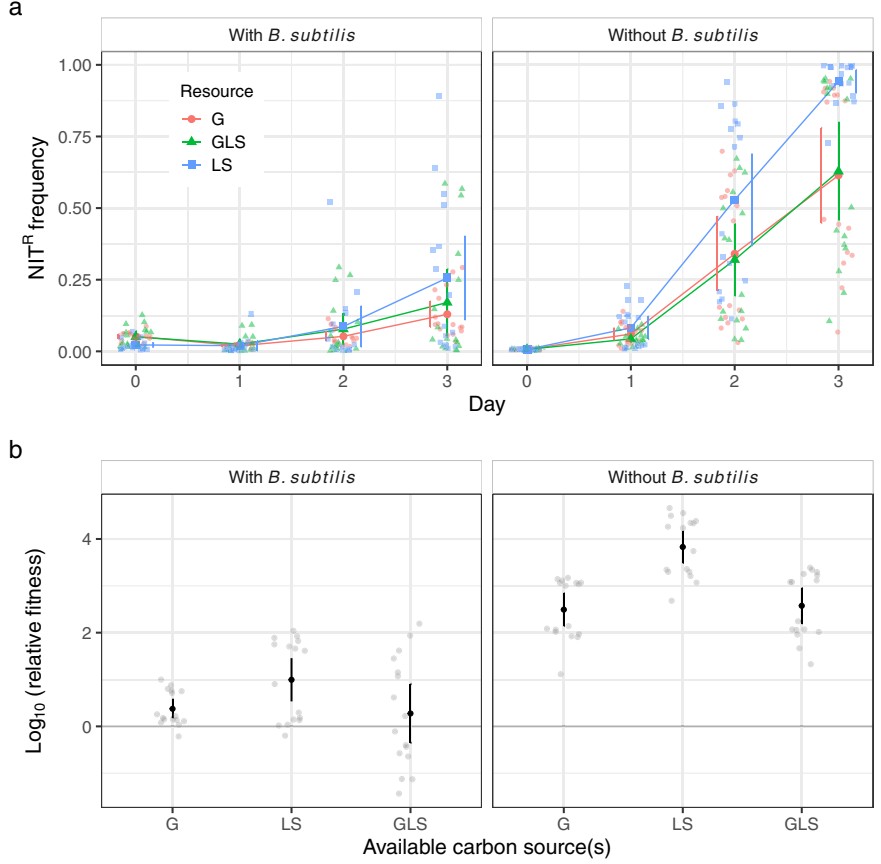

**Fig. 2 Resource competition has no effect on NIT resistance dynamics. a** Lines depict increases in frequencies of NIT[R] *E. coli* cells over 3 days at a quarter of MIC for both panels. Colors depict dynamics at the three available carbon sources in the media (G: 0.2% Glu; LS: 0.2% Lac + 0.3% Salicin; GLS: 0.1% Glu + 0.1% Lac + 0.15% Salicin). **b** Depiction of log-transformed relative fitness of NIT[R] cells in the presence and absence of *B. subtilis* after 3 days of competition with NIT[S] cells in the different carbon source regimes. The log-transformed relative fitness value of zero (gray horizontal line) represents no fitness advantage to either NIT[R] or NIT[S] strains, while positive values indicate an advantage to NIT[R] strains. Lighter points depict individual replicates, darker points represent means, and error bars represent 95% confidence intervals (*t*-distribution, $n = 16$ for both panels).

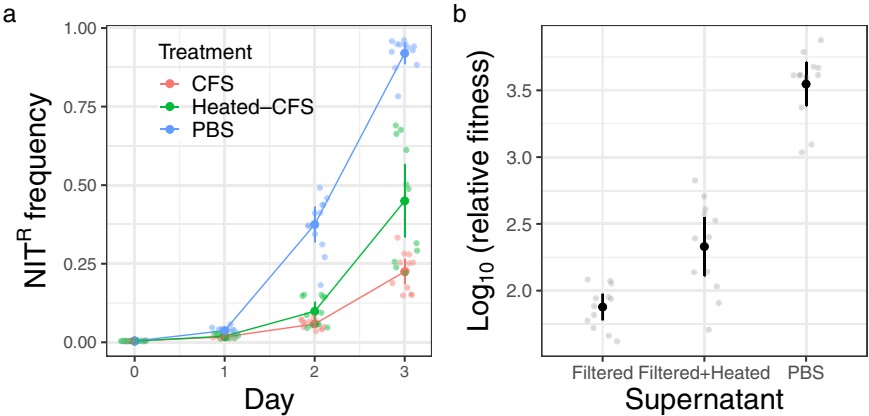

**Fig. 3 Extracellular compounds from *B. subtilis* dampen NIT resistance enrichment. a** Lines depict increase in the frequency of NIT[R] cells over 3 days at sub-MIC. Colors depict dynamics in the presence of either added PBS or cell-free supernatant (CFS) or heat-treated cell-free supernatant. **b** Depiction of log-transformed relative fitness of NIT[R] cells after 3 days of competition with NIT[S] cells under three supernatant treatment regimes. The log-transformed relative fitness value of zero represents no fitness advantage to either NIT[R] or NIT[S] strains, while positive values indicate an advantage to NIT[R] strains. Lighter points depict individual replicates, darker points represent means and error bars represent 95% confidence intervals (*t*-distribution, $n = 12$ for both panels).

(70 nM)). In the presence of NIT, the $NIT^S$ strain showed 3.1–7.8% higher absolute $OD_{600}$, while the $NIT^R$ strain showed a 1.6–5.0% decline in the final $OD_{600}$ values at different concentrations of the protein (Supplementary Fig. 6) when compared to control treatment without YydF. The protein did not exhibit any significant effect on $NIT^R$ and $NIT^S$ growth in the absence of the antibiotic. Overall, these findings are compatible with the notion that YydF dampens $NIT^R$ enrichment.

A previous study has shown that the pre-pro-YydF has to be modified to the epipeptide YydF* by other proteins of the *yyd* operon before it starts affecting *B. subtilis* growth[46]. Accordingly, we sought to determine if the epipeptide YydF* could also differentially affect the growth of $NIT^S$ and $NIT^R$ *E. coli*. However, unlike that seen with pre-pro-YydF, we did not find any difference in log-transformed relative $OD_{600}$ between $NIT^S$ and $NIT^R$ *E. coli* both in the presence and absence of NIT at all three YydF* concentrations (Supplementary Fig. 7, Welch's one-sided *t* test: $t_{2.134} = -2.337$, $P = 0.54$ (5 μM); $t_{2.592} = 0.105$, $P = 0.38$ (2.5 μM) and $t_{3.516} = -2.034$, $P = 0.8$ (1.25 μM)]. No growth was observed in the presence of NIT in both control and test populations at the highest tested YydF* concentration of 10 μM (average $OD_{600}$ after 24 h reached 0.04 and 0.12 in the presence and absence of YydF*, respectively).

Further, YydF* was shown to induce envelope stress in *B. subtilis* and since we found pre-pro-YydF to slightly affect $NIT^R$ *E. coli* growth we looked for evidence of pre-pro-YydF induced

differential transcription of envelope stress-related genes among $NIT^R$ and $NIT^S$ *E. coli*. However, a qPCR analysis did not reveal peptide-induced differential transcription of envelope stress-related genes *cpxA* and *rpoE*[47] among $NIT^R$ and $NIT^S$ *E. coli* strains (Supplementary Fig. 8), suggesting that YydF* has different physiological effects in *B. subtilis* and *E. coli*.

## Discussion

We show that interspecific interaction with *B. subtilis* can significantly dampen sub-MIC selection for NIT resistance in *E. coli* (Fig. 1). Moreover, the slowdown of resistance selection is not linked with resource competition between the two interacting species (Fig. 2). Instead, the significant reduction is probably mediated by extracellular compounds (Fig. 3), most notably the extracellular peptide YydF (Fig. 6).

The social nature of microbial existence is now well established, although most interspecies interactions in these microcosms are proposed to be antagonistic[48]. The extent to which these interactions can determine the outcomes of antibiotic treatment is still a black box awaiting deciphering. This study clearly shows that interspecies interactions can profoundly change the relative fitness of competing resistant and susceptible bacteria in a mixed population[26]. Extracellular molecules produced by bacterial species can modify its environment thus modulating the relative fitness of antibiotic-resistant strains. For example, acidification[25], modification of antibiotics[22,49] and bacteriocin production[50] have been shown to modulate the relative fitness of antibiotic-resistant cells in a population. Previous studies have shown that the cell-free supernatant of a grown *B. subtilis* culture possesses a wide variety of growth inhibitory compounds (such as surfactins, iturin A, fengycin, etc.[51]) as well as a wide variety of proteins and enzymes[52]. Recently, cell-free supernatant of *B. subtilis* was shown to inhibit *Staphylococcus aureus* growth as well as make them susceptible to penicillin and gentamicin by compromising the integrity of its cell wall[53]. Here, we not only find that cell-free supernatant of *B. subtilis* to be inhibitory to *E. coli* growth (Fig. 4) but also show that an excreted extracellular peptide can be an important driver in reducing the relative fitness of $NIT^R$ *E. coli* (Fig. 6).

The epipeptide YydF* (recently renamed to EpeX*[54]) has been shown to carry antibacterial activity towards *B. subtilis* itself[46] by depolarizing the cell membrane through membrane permeabilization[45]. Moreover, the peptide is synthesized by the *yyd* operon (recently renamed to *epe* operon[54]), the expression of which is driven by a $σ^A$-dependent promoter[55]. The peptide is produced at the onset of stationary phase and elicits cell envelope stress in *B. subtilis* cells through the LiaRS two-component system[45]. Here, we show that pre-pro-YydF seems to have a small differential effect on the growth of $NIT^R$ *E. coli* in the presence of NIT (Supplementary Fig. 6), which could partly explain its role in the dampening of $NIT^R$ enrichment at sub-MIC (Fig. 6).

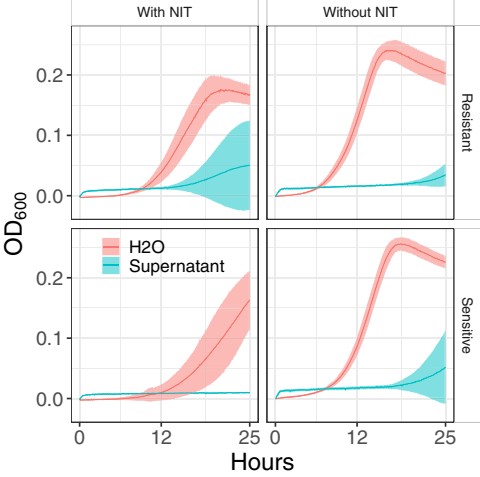

**Fig. 4 *B. subtilis* supernatant inhibits *E. coli* growth in minimal media.** Growth of $NIT^S$ and $NIT^R$ *E. coli* depicted as change in $OD_{600}$ over 24 h in minimal media diluted with either sterile water or cell-free supernatant from *B. subtilis*. Each line is an average from three independent biological replicates. Error bars, shown as ribbons, represent 95% confidence intervals (*t*-distribution, $n = 3$).

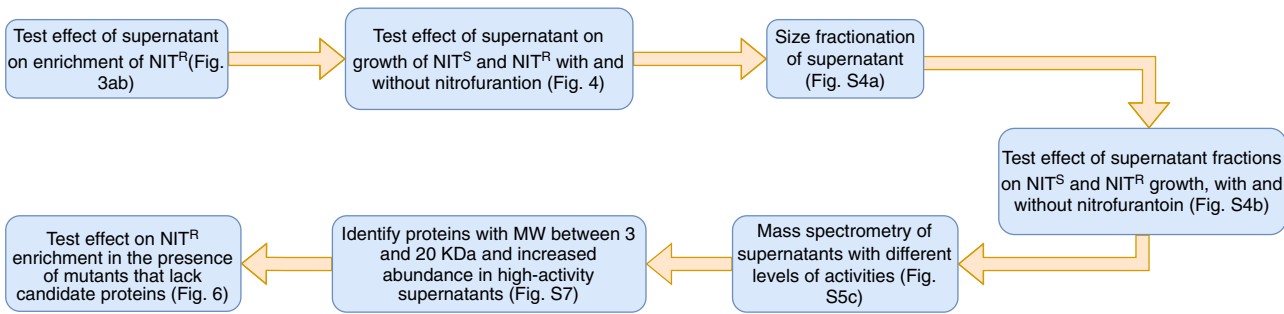

**Fig. 5 Identification of YydF and YorD.** Flow chart explaining the method used to identify and test proteins or peptides that has a significant effect on *E. coli* NIT resistance dynamics.

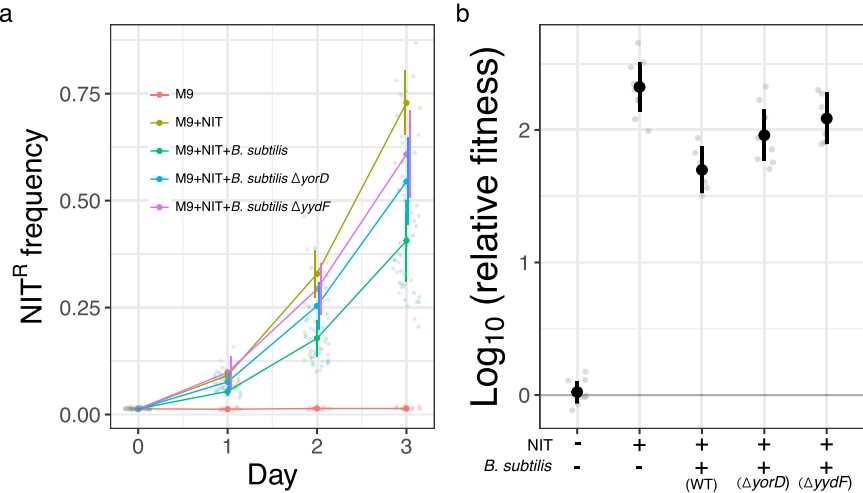

**Fig. 6 Deletion of *yydF* gene from *B. subtilis* 168 offsets dampening of NIT resistance enrichment. a** Lines depict increases in frequencies of NIT[R] cells over 3 days at sub-MIC. Colors depict dynamics during interspecies interactions with different *B. subtilis* strains. The wild-type strain used here is the parent strain of Δ*yydF* and Δ*yorD* ordered from BGSC (see "Methods"). **b** Depiction of differences in log-transformed relative fitness of NIT[R] cells after 3 days of competition with NIT[S] cells in the presence of *B. subtilis* strains deleted for or carrying functional copies of either *yydF* or *yorD* genes. The log-transformed relative fitness value of zero (gray horizontal line) represents no fitness advantage to either NIT[R] or NIT[S] strains, while positive values indicate an advantage to NIT[R] strains. All competitions were conducted in M9-Glucose with or without NIT, as mentioned in the legends (**a**) or on the x axis (**b**). Lighter points depict individual replicates, darker points represent means and error bars represent 95% confidence intervals (*t*-distribution, *n* = 8 for both panels).

However, the epipeptide YydF* does not seem to have any differential effect on NIT[R] *E. coli* at the concentrations we tested (Supplementary Fig. 7). In addition, the lack of growth in control populations at the highest concentration of the peptide restricts our ability to conclude anything about the role of YydF* in dampening NIT[R] enrichment, if any. It remains unclear how the peptide affects *E. coli*, it might be acting on all *E. coli* cells in the population to reduce growth, but it could also be specifically targeting cells that lack nitroreductases NfsA and NfsB, especially since the general reduction in the number of *E. coli* generations (through dilution of growth media, Supplementary Fig. 2) still lead to a dampening of NIT[R] enrichment. If the specific activity of the peptide is directed at only NIT[R] cells this could be a novel way of targeting NIT resistance. However, it should be noted that the activity of YydF does not fully explain the change in dynamics of NIT resistance and interactions with other molecular components (such as YorD) could also play a vital role in the overall dampening of NIT[R] enrichment by *B. subtilis* (Fig. 6). Future work will determine the nature of *E. coli* interactions with pre-pro-YydF as well as YydF* epipeptide and whether these interactions can largely explain the dampening of NIT[R] enrichment by *B. subtilis*.

Studies employing synthetic multispecies microbial systems are rare, especially those involving the study of evolutionary dynamics. For example, utilization of synthetic microbial biosystems has revealed modulation in secondary metabolite responses[56], anti-predator strategies among bacterial prey[57], traits which are important for the resilience of microbial communities[58] and factors driving the diversification of CRISPR immunity[59], emphasizing the importance of such systems in understanding the mechanisms of interspecies microbial interactions that play a role in modulation of bacterial traits, including antibiotic resistance. Our finding that the interaction between two bacterial species, powered by an extracellular peptide, could impact on the evolution of antibiotic resistance at sub-MIC should help in the endeavor of predicting the trajectory of a resistance trait once it has emerged in a population. Further, identifying the molecular drivers of interspecies interactions affecting the evolutionary dynamics of resistant traits should aid

us in better controlling resistance emergence and evolution in bacterial populations.

## Methods

**Strains and media**. *Escherichia coli MG1655* strains (DA28100 and DA28102, both derived from parent strain DA4201) previously reported to carry chromosomal copies of fluorescent protein genes *bfp* and *yfp* were used for all experiments[6]. Fluorescent nitrofurantoin-resistant strains were constructed by moving the fluorescent protein genes from DA28100 and DA28102 into nitrofurantoin-resistant DA65117 (Δ*nfsAB*)[60] using the P1 transduction protocol with chloramphenicol marker for selection[61]. The cultures were streaked onto LB agar (5 g yeast extract, 10 g Tryptone, 10 g NaCl, and 15 g agar per liter) plates and individual colonies were then inoculated into sterile liquid minimal media (1× M9 salts, 1 mM MgSO$_4$, 0.1 mM CaCl$_2$, 50 mg/L Tryptophan, 10 μM MnSO$_4$, 1 μM FeSO$_4$) with 0.2% glucose for overnight growth. *B. subtilis* subsp. *subtilis* 168 (also known as *B. subtilis* 168, a kind donation from Andreas Porse from Danmarks Tekniske Universitet) was used for most interspecies interaction experiments. *B. subtilis* Δ*yydf* (trpC2 Δ*yydF*:kan; BGSCID: BKE40180) and Δ*yorD* (trpC2 Δ*yorD*:kan; BGSCID: BKE20420) strains were ordered from Bacillus Genetic Stock Center (BGSC), Columbus, Ohio, the US. For competitions involving knockout strains from BGSC, the BGSC parent strain (BGSCID: 1A1) was used as wild-type.

**Competitions**. Competition experiments to determine nitrofurantoin resistance enrichment as well as to estimate relative fitness were performed using the genetically tagged strains with chromosomal copies of *bfp* and *yfp* as described above. Six to eight colonies of each bacterial strain were grown in minimal media with 0.2% glucose. Then the susceptible strains tagged with one of the two fluorescent markers were mixed at 99:1 with the nitrofurantoin-resistant strain carrying the other marker in the same media with sub-MIC levels of nitrofurantoin (a quarter of MIC or unless otherwise mentioned). One microliter of the *E. coli* culture mix was then mixed with an equal volume of sterile water or fully grown *B. subtilis* culture (grown the same way as *E. coli* cultures) before being added to a well in a 96-well plate containing 198 μL of sterile minimal media to start the competitions. The competitions were performed under shaking at 32 °C. The following day, the competing strains were passaged by 100-fold dilution into fresh medium and were allowed to grow for another 24 h and passaged once more the next day. The ratio of resistance to susceptible cells was measured by counting at least 10$^5$ cells using a fluorescence-activated cell sorter (BD FACS Aria) before the passage and every 24-h thereafter for 4 days in total. The relative fitness of the resistant strain was estimated using the formula of Ross-Gillespie et al.[43]:

$$v = \frac{x2(1-x1)}{x1(1-x2)} \qquad (1)$$

where *x*1 and *x*2 are the initial and final frequencies (after three rounds of competition) of the NIT[R] cells, respectively.

Thus, the relative fitness estimate uses the initial and final frequencies of the NIT$^R$ cells and does not generate per generation or per unit time estimate. The frequency of *B. subtilis* is estimated as the ratio of the total number of non-fluorescent cells over the total number of fluorescent cells in each sample where *B. subtilis* was added on day 0. The frequency of resistant cells (within *E. coli* populations), *B. subtilis* cells (in total population) as well as relative fitness was averaged across all 12–16 independent biological replicates across the two marker pairs.

**Growth measurements**. MIC using broth microdilution methods as well as growth of parental and fluorescent *E. coli* populations were determined in minimal medium using a Bioscreen C reader (Oy Growth Curves Ab Ltd). Three independent cultures for each strain were grown overnight and diluted 1:100 in fresh minimal medium. Three hundred microlitres of this suspension was added to the wells of the honeycomb plates. The plates were incubated in the Bioscreen C analyzer at 32 °C with shaking for 24–40 h as appropriate. The OD at 600 nm wavelength ($OD_{600}$) value was measured every 4 min.

**Experiments with supernatant**. *B. subtilis* frozen stocks were streaked onto fresh LB agar plates and incubated overnight at 32 °C. Individual colonies were then inoculated into 25 mL minimal media with 0.2% glucose in 50 mL flasks and grown for 25 h under shaking at 32 °C. Fully grown cultures were then centrifuged at 4500 rpm for 15 min, and the supernatant was then filtered through a 0.2-micron filter to get rid of any cells. Sterile minimal media was then mixed with an equal volume of either the supernatant or with PBS (for the initial experiment) or water (for subsequent experiments) and used as growth media for competitions at sub-MIC of NIT. To test the effect of supernatants on *E. coli* growth, sterile minimal media with 0.2% glucose were mixed with an equal volume of supernatant or water and $OD_{600}$ was measured every 4 min using Bioscreen under constant shaking at 32 °C. Non-fluorescent parental strains (DA4201 and DA65117) were used to test for the effect of the supernatant on growth. Heat-treated supernatant was made by heating the supernatant in an autoclave at 121 °C for 20 min.

**Supernatant fractionation**. Supernatant from *B. subtilis* were fractionated using Pierce$^{TM}$ Protein Concentrators from ThermoFisher with filters at desired molecular weight cut-offs (MWCO). The supernatant was first passed using a centrifuge at $4000 \times g$ through a column with MWCO of 10 KDa for 60 min, the retentate formed fraction 1 with components >10 KDa. The filtrate from the previous filtration was then filtered through a column with MWCO of 3 KDa at $4000 \times g$ for 60 min, the retentate of which formed fraction 2 with components between 3 and 20 KDa. Fractions 1 and 2 were reconstituted to a final volume of 25 mL with fresh sterile minimal media and filtered through the same column again to rid of acids or other small molecular compounds. The resulting fractions were then reconstituted to a final volume of 25 mL with fresh sterile minimal media before being used to test their effect on *E. coli* growth the same way as with supernatant. The supernatant and the fractions were generated fresh for each experiment and were stored at 4 °C between use.

**Proteomics**. An aliquot of frozen *B. subtilis* 168 stock was thawed and spread onto multiple fresh LB agar plated with a sterile loop and incubated overnight at 32 °C. More than 20 equally sized colonies were then resuspended in 1 mL of minimal media and 900 μL of which was inoculated into 25 mL minimal media in 50-mL flasks and grown for exactly 25 h under shaking at 32 °C and used to harvest supernatant as before and tested for their effect on *E. coli* growth. Supernatant from replicate *B. subtilis* cultures had a varied non-repeatable effect on *E. coli* growth, so from the more than 20 cultures that were tested, we chose three cultures that showed high, intermediate and low effect on *E. coli* growth. Fresh supernatants were harvested from the three cultures (two replicates of each culture) as before, but the effect of the supernatant on growth was not repeatable, however, the three cultures still have varied effect on *E. coli* growth (Supplementary Fig. 3). Supernatants from both replicates of the three cultures were then concentrated using a Pierce$^{TM}$ Protein Concentrators with MWCO of 3 KDa, protein bands in the supernatant were checked using SDS-PAGE (Supplementary Fig. 9) and were then stored at −80 °C until proteomic analysis. The concentrated supernatant samples were sent to Proteomics Core Facility at the University of Gothenburg for mass spectrometry-based analysis. Mass spectrometry-based relative quantification of protein across the samples were done using tandem mass tag (TMT) technology[62]. The analysis detected 346 unique protein or peptide sequences (Supplementary Data 1) across the six samples from which the peptides of interest were identified by first filtering for proteins or peptides that were relatively abundant in high-activity supernatant (1 S) as compared to the one with lowest activity (9 S) [(9 S)/(1 S) abundance ratio <0.1]. Next, we filtered the remaining proteins for intermediate presence in 1 S relative to the supernatant of intermediate activity (4 S) [(4 S)/(1 S) abundance ratio between 0.2 and 0.7]. Finally, we filtered out the proteins and peptides with MW higher than 20 KDa to arrive at five peptides (Supplementary Fig. 10) from which two (YydF and YorD) were characterized to have activities of interest as per the protein database Uniprot and were hence considered for further analysis.

**Experiments with pre-pro-YydF and YydF\***. YydF peptide or pre-pro-YydF was synthesized and purified by the Protein Science Facility of the Karolinska Institutet

in Stockholm, Sweden. Briefly, based on the available genetic sequence of *yydF*, the peptide was transformed in to *E. coli* BL21 with a SUMO tag. The cells were cultivated in Terrific Broth (TB) medium. Protein expression was induced with isopropyl-β-D-1- thiogalactopyranoside (IPTG) and the protein was purified by immobilized metal-ion chromatography (IMAC), followed by size exclusion chromatography (SEC). The SUMO tag of the peptide was cleaved with Ubiquitin-like-specific protease 1 (ULP1). The cleaved SUMO tag was removed by reverse IMAC purification. The peptide was delivered in buffer at a concentration of 0.8 mg/mL (or 140 μM) in 20 mM HEPES, 300 mM NaCl, 10% glycerol, 2 mM TCEP, pH 7.5. This was diluted in sterile minimal media to appropriate concentrations before being checked for effect on *E. coli* growth.

The epipeptide YydF\*: WYFV$^D$KSKENRWI$^D$LGSGH (where "D" denotes D-amino acid residues) was synthesized by Red Glead Discovery AB based at Lund, Sweden, as per the sequence reported in ref.[45]. The lyophilized peptide was dissolved in 25% acetic acid and then added to minimal media to check for its effect on *E. coli* growth. Dissolved peptide was stored at −80 °C between uses, peptide was aliquoted into a number of tubes to avoid repeated thawing and freezing. Growth in the presence of peptide were compared with control treatments that contained equivalent amount of acetic acid in minimal media.

**Measurement of changes in expression of *cpxA* and *rpoE***. Overnight cultures of NIT$^S$ and NIT$^R$ *E. coli* were diluted 1:100 in 20 mL minimal media and were grown at 32 °C to $OD_{600}$ value of 0.5, upon which pre-pro-YydF were added to achieve a final concentration of 70 nM and grown for further 30 min. The entire culture was then spun down and resuspended in 1 mL of RNA protect reagent, vortexed, kept on ice for 5 min, spun down again and the pellets were stored at −80 °C until RNA extraction. RNA extraction and cDNA production was performed as per ref.[63] and described herein. RNA extraction was performed using the RNeasy Mini Kit (Qiagen) as per the manufacturer's protocol. The extracted RNA was then treated with DNase using the Turbo DNA-free kit (Ambion) as per the manufacturer's protocol. About 500 ng of RNA (quantified using the Qubit RNA BR assay kit) was used for cDNA preparation using the High Capacity Reverse Transcription Kit (Applied Biosystems). PerfeCta Sybr Green SuperMix (Quanta Biosciences) was used to perform the RT-qPCRs. The transcript abundance measured for the different genes was normalized to the geometrical mean of the levels obtained for housekeeping genes *cysG*, *rssA*, and *hcaT*. Ratio of expression in the presence of the peptide over that in the absence was used to generate plots. Two biological and three technical replicates were used in each case.

**Statistics and reproducibility**. Relative fitness values in all cases were log-transformed to achieve normal distribution. The difference in average relative fitness values post transformation in the presence and absence of *B. subtilis* was tested using one-sided *t* tests (Fig. 1b). The effect of available carbon source (Fig. 2b), supernatant treatment (Fig. 3b), and different *B. subtilis* strains (Fig. 6b) on relative fitness was tested using ANOVA and when found significant was further tested with Tukey's HSD posthoc test to determine if there are significant differences in average log-transformed relative fitness between the treatments. Statistical significance of the effect of pre-pro-YydF and YydF\* on growth of susceptible and resistant strains in the presence and absence of NIT was tested using one-sided *t* tests with alpha set at 0.0125 following Bonferroni's multiple-significance-test correction (Supplementary Figs. 6 and 7). The number of biological replicates for each experiment is reported as *n* in each of the respective figure legends.

Statistical analyses were performed in R[64] using RStudio 2022.02.3 + 492. All graphs were made using the R package ggplot2[65] and combined using the package patchwork[66]. The flow diagram (Fig. 5) was made using draw.io (www.diagrams.net).

**Reporting summary**. Further information on research design is available in the Nature Portfolio Reporting Summary linked to this article.

## Data availability

Data used for all the figures in the manuscript is available on Figshare (https://doi.org/10.6084/m9.figshare.21977045.v1). The mass spectrometry proteomics data have been deposited to the ProteomeXchange Consortium via the PRIDE[67] partner repository with the dataset identifier PXD040869.

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

## Acknowledgements

This work was supported by grant UPD2020-0072 from the Wenner-Gren Foundation (RRN), grant KAW 2018.0168 from the Wallenberg Foundation (DIA) and grant 2021-02091 from the Swedish Research Council, Medicine and Health (DIA). Additionally, we thank all current and former members of the Andersson lab as well as other scientists in D7:3 corridor and Per Jemth for helpful discussions. We thank Omar Mahmud Warsi for the discussions as well as for his help in the lab. Marie Wrande and Po-Cheng Tang for help with flow cytometry. Ulrika Lustig for help in the lab. Members of Samay Pande lab at IISC Bangalore and Deepa Agashe lab at NCBS Bangalore for inputs. Emilia Strandback and Tomas Nyman at Protein Science Facility, KI, Stockholm, for help with YydF peptide, Egor Vorontsov and Carina Sihlbom at Proteomics Core Facility, University of Gothenburg, for help with supernatant analysis and Samara Mamidi at Red Glead Discovery for help with YydF* epipeptide.

## Author contributions

R.R.N. and D.I.A. planned the project and co-wrote the final manuscript. R.R.N. performed all the experiments, collected and analyzed the data, and wrote the initial draft.

## Funding

## Competing interests

The authors declare no competing interests.
