## [Peer Review File · Communications Biology]

Reviewers' comments:

Reviewer #1 (Remarks to the Author):

Review on "Interspecies interaction reduces selection for antibiotic resistance in *Escherichia coli*" by Nair and Andersson

In this manuscript the authors tackle with a very important and timely topic: how antibiotic resistance develops in multispecies communities. In their experimental work they use two species synthetic community containing *B. subtilis* and two genotypes of *E. coli*.

The main finding is that the presence of *B. subtilis* dampens NIT resistance enrichment in *E. coli* and this is mediated through an extracellular compound produced by the *B. subtilis* and not direct ecological interaction via resource competition. This makes results even more interesting.

I really enjoyed reading this manuscript. Topic is relevant, experiments are carried out carefully and conclusions are backed up with relevant data. Overall quality of the manuscript is very high. I think this would be a very nice addition to the literature and should be published in the *Communications in Biology*.

Only one very minor comment:

"Evolution of microbial traits depend on the interaction of a species with its environment as well as with other coinhabiting species, including predators. "

"Evolution of microbial traits depend on the interaction of a species with its environment as well as ecological interaction with other species. "

I would not mention predators to be less specific at this point.

Reviewer #2 (Remarks to the Author):

Although the evolution of antimicrobial resistance has received a lot of experimental attention in recent years, there are few studies that explicitly take into account the biotic context in which bacterial evolution invariably takes place. Bacteria live in dense and complex communities and engage in many different types of interactions that could lead to physiological and genetic changes that in turn could affect resistance evolution. A few studies have investigated competition of resistant (R) and susceptible (S) cells in complex communities. Although arguably most realistic, such experimental designs are also less tractable. The current paper describes a very simple system consisting of three strains, focal R and S *E. coli* and *B. subtilis*, and the prodrug nitrofurantoin to investigate how the presence of a second species could affect the relative fitness of S and R strains at subMIC concentrations.

The presence of *Bs* has a negative effect on selection for R *Ec* across the subMIC range used. This is not due to resource competition as shown by eliminating different carbon compounds that can be differentially used by the two species. Instead secreted compounds, specifically those that cannot withstand heating, are shown to be responsible for this effect. This indicates the action of denaturable proteins having a differential fitness effect on S and R *Ec*. Fractionation and comparison of replicates with varying effects on inhibition indicated the effect was due to possibly two proteins; subsequent competition experiments using two knockout mutants demonstrated that the protein YydF was responsible, and only so in the presence of the antibiotic. The protein was shown to not inhibit *Ec* as it does in *Bs* through envelope stress.

In short, this paper describes a proof-of-principle experiment demonstrating that species interactions can influence the trajectory of invasion of resistant cells (proof of principle: interactions with different species are likely to be highly varied and it is conceivable many bacterial interactions do not have an effect on resistance evolution of a second species). The authors have managed to narrow down the genetic (albeit not the functional) basis of this interaction as well. The paper is well-written and the structure is clear. I am convinced by the methods and data analysis. I only have a few minor comments pasted below.

Abstract "including predators." Why mention predators specifically among all the different types of interactions, when this is not what is studied in your paper?

L22: "which should negatively affect population density, reducing the number of possible doublings of the resistant strain thereby influencing its evolution in the presence of antibiotics." But it also reduces the doublings of the sensitive strain which would offset this to some degree. 'evolution' here is a bit vague: does this refer to resistance evolution (the strain is already resistant).

L26 this is unclear to me "presence of another species in the ecosystem might result in sharing of the antibiotics present in the environment,"

Fig. 1b is not mentioned in the Results section.

Fig 4. It is not clear to me whether the lines represent relative growth rate or growth of only S or R (and if so which). Source data is made available: where?

Fig 6A: the legend mentions different constructs but also media or absence of antibiotics, I find this not 100% clear, eg. I presume 'no NIT' is not in M9 but do not know for sure.

Reviewer #3 (Remarks to the Author):

The authors present data indicating that direct competition by *B. subtilis* can result in increased inhibition of NIT resistant *E. coli* compared to NIT sensitive strains. As a result, selection for resistance at sub-inhibitory concentrations is suppressed in the presence of *B. subtilis*. The authors propose that the peptide YydF produced by *B. subtilis* has a differential effect on resistant and sensitive *E. coli*, and contributes to the observed selection dynamic. The experiments are well done but I do have some questions about the data, relevance, and interpretation of the results.

My main comment is it is unclear to me why YydF would produce differential inhibition between resistant and sensitive *e. coli*. Is it due to additive effects of cost of loss of function of NfsA and NfsB or is there some other interaction occurring? I would like to have seen more investigation into what is driving this effect.

I understand the rationale of investigating sub-MIC levels of NIT, but I would have liked to have seen data on what concentration of antibiotic is required to positively select for NIT resistance in the presence of *B. subtilis*. This data may also give some insight into how *B. subtilis* is preventing the selection for resistance.

Line 43 – this seems a bit of a stretch; is there any evidence of coinfection, particularly in UTI infections, where the interactions and their effects on selection for NIT resistance could have an effect?

Line 56 – does *B. subtilis* have the same suppressive effect if the initial frequency of resistance was higher? I.e. does *B. subtilis* only prevent resistance from invading from rare or can it suppress selection when starting from equal fitness – is the effect frequency dependent?

Figure 1a – What are the coculture dynamics? Does *B. subtilis* survive/outcompete both *E. coli* strains during the experiments? It would be useful to display relative density of *E. coli*/*B. subtilis* to help interpret the results.

Figure 1b – is 0 equal fitness? It would be useful to have a horizontal line to represent this in all figures presenting relative fitness.

Line 110 – "before coming out of *B. subtilis* cells" feels very unscientific.

Figure 4 – there are only two lines in each panel – is it really necessary to remove error bars for

clarity? Or they are very large? This is important information that the readers should be shown.

The data presented in figure 6 shows the resistance increasing from 0.01% to ~37% in the presence of *B. subtilis*, this is very different from the data presented in Figure 1 where the frequency of resistance does not change at all. Why is the effect so different between the two figures? Is the effect presented in figure 1 repeatable?

Response to reviewer's comments for the paper entitled “Interspecies interaction reduces selection for antibiotic resistance in *Escherichia coli*”

Reviewer #1 (Remarks to the Author):

We thank the reviewer for their careful reading, positive endorsement and thoughtful suggestions to improve the manuscript.

*Review on “Interspecies interaction reduces selection for antibiotic resistance in *Escherichia coli*” by Nair and Andersson*

*In this manuscript the authors tackle with a very important and timely topic: how antibiotic resistance develops in multispecies communities. In their experimental work they use two species synthetic community containing *B. subtilis* and two genotypes of *E. coli*.*

*The main finding is that the presence of *B. subtilis* dampens NIT resistance enrichment in *E. coli* and this is mediated through extracellular compounds produced by the *B. subtilis* and not direct ecological interaction via resource competition. This makes results even more interesting.*

*I really enjoyed reading this manuscript. Topic is relevant, experiments are carried out carefully and conclusions are backed up with relevant data. Overall quality of the manuscript is very high. I think this would be a very nice addition to the literature and should be published in the *Communications biology*.*

We thank the reviewer for the recommendation to publish.

Only one very minor comment:

“Evolution of microbial traits depend on the interaction of a species with its environment as well as with other coinhabiting species, including predators. ”

“Evolution of microbial traits depend on the interaction of a species with its environment as well as ecological interaction with other species. ”

I would not mention predators to be less specific at this point.

We agree with the reviewer and have removed the reference to predators.

Reviewer #2 (Remarks to the Author):

We thank the reviewer for their careful reading and thoughtful suggestions to improve the manuscript.

Although the evolution of antimicrobial resistance has received a lot of experimental attention in recent years, there are few studies that explicitly take into account the

biotic context in which bacterial evolution invariable takes place. Bacteria love in dense and complex communities and engage in many different types of interactions that could lead to physiological and genetic changes that in turn could affect resistance evolution. A few studies have investigated competition of resistant (R) and susceptible (S) cells in complex communities. Although arguably most realistic, such experimental designs are also less tractable. The current paper describes a very simple system consisting of three strains, focal R and S E. coli and B. subtilis, and the prodrug nitrofurantoin to investigate how the presence of a second species could affect the relative fitness of S and R strains at subMIC concentrations. The presence of Bs has a negative effect on selection for R Ec across the subMIC range used. This is not due to resource competition as shown by eliminating different carbon compounds that can be differentially used by the two species. Instead secreted compounds, specifically those that cannot withstand heating, are shown to be responsible for this effect. His indicates the action of denaturable proteins having a differential fitness effect on S and R Ec. Fractionation and comparison of replicates with varying effects on inhibition indicated the effect was due to possibly two proteins; subsequent competition expts using two knockout mutants demonstrated that the protein YydF was responsible, and only so in the presence of the antibiotic. The protein was shown to not inhibit Ec as it does in Bs through envelope stress. In short, this paper describes a proof-of-principle experiment demonstrating that species interactions can influence the trajectory of invasion of resistant cells (proof of principle: interactions with different species are likely to be highly varied and it is conceivable many bacterial interactions do not have an effect on resistance evolution of a second species). The authors have managed to narrow down the genetic (albeit not the functional) basis of this interaction as well. The paper is well-written and the structure is clear. I am convinced by the methods and data analysis. I only have a few minor comments pasted below.

Abstract “including predators.” Why mention predators specifically among all the different types of interactions, when this is not what is studied in your paper?

We agree with the reviewer and we have removed the reference to predators.

L22: “which should negatively affect population density, reducing the number of possible doublings of the resistant strain thereby influencing its evolution in the presence of antibiotics.” But is also reduces the doublings of the sensitive strain which would offset this to some degree. ‘evolution’ here is a bit vague: does this refer to resistance evolution (the strain is already resistant).

If a strain is known to be relatively fitter than its counterpart (as NIT^R strains are in this case) then they increase their proportion in the population at the end of every generation, simply because they are able to undergo more doublings in the given environment and if the number of doublings of both strains is restricted in any way (such as due to presence of a second species) then that might automatically decrease the enrichment rate of the fitter strain. That is the logic behind the sentence. We have tried to clarify it further.

Here, the term “evolution” is meant in a classic sense, i.e. as the change in allelic frequency in a population. The enrichment of a particular trait is thus a process of evolution for that trait in that population. We believe the author is referring to the process of *de novo* emergence of a particular trait in a population, which generally happens before evolution. We believe the use of the word “evolution” here is appropriate, however, to avoid confusion we have replaced the word with “enrichment”.

L26 this is unclear to me “presence of another species in the ecosystem might result in sharing of the antibiotics present in the environment,”

The presence of another species can reduce the free concentration of antibiotics in the environment (e.g. by sequestering the drug) which would have otherwise interacted with the focal species, this is what we meant by sharing antibiotics. We have slightly changed this sentence to clarify this.

Fig. 1b is not mentioned in the Results section.

Thank you for pointing this out, we have now included a reference to figure 1b in the results section.

Fig 4. It is not clear to me whether the lines represent relative growth rate or growth of only S or R (and if so which). Source data is made available: where?

We apologise for any confusion the figure may have caused. The lines here are growth curves representing the change in optical density in the presence of supernatant or sterile water and not the growth rate or relative growth rate. Growth of both S (bottom panels) and R (top panels) are reported in the figure in the presence (left panels) and absence (right panels) of nitrofurantoin.

Source data is generally made available post-acceptance and prior to publication of the manuscript, but we acknowledge that the reviewer might need a closer look at the data, especially in the absence of error bars so we have included error bars in the figure and have included source data for all the figures.

Fig 6A: the legend mentions different constructs but also media or absence of antibiotics, I find this not 100% clear, eg. I presume ‘no NIT’ is not in M9 but do not know for sure.

We apologise for the confusion and thank the reviewer for pointing this out. We have now changed the legends in the figure to “M9+NIT” and “M9” and hope this avoids confusion.

Reviewer #3 (Remarks to the Author):

We thank the reviewer for their careful reading and thoughtful suggestion to improve the manuscript.

The authors present data indicating that direct competition by B. subtilis can result in increased inhibition of NIT resistant E. coli compared to NIT sensitive strains. As a result, selection for resistance at sub-inhibitory concentrations is suppressed in the presence of B. subtilis. The authors propose that the peptide YydF produced by B. subtilis has a differential effect on resistant and sensitive E. coli, and contributes to the observed selection dynamic. The experiments are well done but I do have some questions about the data, relevance, and interpretation of the results.

My main comment is it is unclear to me why YydF would produce differential inhibition between resistant and sensitive e. coli. Is it due to additive effects of cost of loss of function of NfsA and NfsB or is there some other interaction occurring? I would have like to have seen more investigation into what is driving this effect.

We fully agree with the reviewer that this is an important question that needs to be understood. To that end, we tried to look at the effect of pre-pro-YydF and the epeptide, YydF*, on the growth of *E. coli* as noted in supplementary figures 7 and 8 (changed to 8 and 9 after revisions) but we did not find a significant effect at the concentrations tested. Additionally, the epeptide was found to be insoluble in water and when dissolved in DMSO, it would readily precipitate in the growth medium even at very low concentrations. We also tried dissolving it in acetic acid, which itself was found to be inhibitory to *E. coli* growth when added to the media. This restricted our ability to work with the peptide at higher concentrations where growth was affected. We tried to see the effect of pre-pro-YydF on mRNA expression of genes related to envelope stress in *E. coli* (since YydF* causes envelope stress in *B. subtilis*), but did not find any evidence for the differential effect on the sensitive and resistant strains (Supplementary figure 9 (now figure 10)).

In light of this, the best way forward would perhaps have been to try to isolate mutants that are resistant to the growth inhibitory effect of the peptides but since they are only reducing growth rates (i.e., they are not lethal), mutant selections would have been challenging. Furthermore, we do not know the exact mode of action of the antibiotic (it is a prodrug that is converted into a free radical that interacts with many types of molecules in the cell) or the physiological consequences of NfsAB inactivation which makes it very difficult to sort out exactly what YydF does to the *E. coli* cell. Thus, the differential activity of the peptide on NIT resistant and sensitive *E. coli* has to be left as an open question as of now and hopefully will be resolved in the future when more information about the antibiotic action, effects of NfsAB inactivation and YydF function will come to light. We hope the reviewer understands these considerations and acknowledges the effort we have already invested in addressing this question.

I understand the rational of investigating sub-MIC levels of NIT, but I would have liked to have seen data on what concentration of antibiotic is required to positively select for NIT resistance in the presence of B. subtilis. This data may also give some insight into how B. subtilis is preventing the selection for resistance.

NIT resistant *E. coli* is getting enriched in all the conditions where the antibiotic is added, the presence of *B. subtilis* simply reduces its relative fitness against the sensitive strain. Even in figure 1, in conditions where antibiotic is present, the log-transformed relative fitness of the resistance strain remains significantly higher than 0 (no fitness advantage to the resistant strain). However, we acknowledge that this is not made clear in the current version of the figure and have added a horizontal line highlighting the point of no fitness advantage to make clear that resistant strains continue to be relatively fitter than the sensitive strain (albeit to a lower extent).

We have conducted this experiment at higher concentrations of the antibiotics and the results of which can be seen below (Figure R1). We have not included this data and figure in the current version of the manuscript, because we believe it does not aid in the interpretation of the data currently included in the manuscript or support any of the conclusions.

Figure R1. Figure representing enrichment dynamics of NIT^R strain at NIT concentrations higher than those reported in the manuscript. Please note that at high NIT concentrations, the growth of *B. subtilis* is also severely inhibited, which confounds any conclusions about interspecies interactions we could derive from these experiments. Error bars represent 95% confidence intervals (*t*-distribution, *n* = 8).

Line 43 – this seems a bit of a stretch; is there any evidence of coinfection, particularly in UTI infections, where the interactions and their effects on selection for NIT resistance could have an effect?

We agree this might be a bit of a stretch, but we would like to keep this remark in the manuscript since it is a reasonable assumption to make considering the number of different types of infections *B. subtilis* can cause in immunocompromised patients. Though we have not found any literature specifically mentioning coinfection the fact *B. subtilis* causes multiple different infections and is consumed as a probiotic and *E. coli* being a human commensal implies possible coinfection and interaction. We are open to removing it if the reviewer insists.

Line 56 – does B. subtilis have the same suppressive effect if the initial frequency of resistance was higher? I.e. does B. subtilis only prevent resistance from invading from rare or can it suppress selection when starting from equal fitness – is the effect frequency dependent?

We agree that the frequency dependence of competition between resistant and sensitive strains in the context of interspecies interactions is indeed a very interesting question. However, the focus of the current manuscript is to study the enrichment of antibiotic resistance starting from a minority (since most antibiotic resistance traits that arise due to *de novo* mutation or migration start from a low frequency in a population) in the presence of interspecies interactions. We then sought to look at molecular determinants that result in the observed change in enrichment dynamics and tried to look at the mechanism of action of the molecule. Thus, we believe that the question of frequency dependence does not fit this manuscript, but will indeed be a very interesting investigation for a separate study.

Figure 1a – What are the coculture dynamics? Does B. subtilis survive/outcompete both E. coli strains during the experiments? It would be useful to display relative density of E.coli/B.subtilis to help interpret the results.

We do have the data from the flow cytometer for the non-fluorescent *B. subtilis* used in the experiments and they are maintained stably during the course of the experiment. We have included the data as a supplementary figure (Supplementary Information Fig. 2 in the revised version).

Figure 1b – is 0 equal fitness? It would be useful to have a horizontal line to represent this in all figures presenting relative fitness.

We apologise for the confusion this has caused and thank the reviewer for pointing this out. We have included a horizontal line in all the relevant figures.

Line 110 – “before coming out of B. subtilis cells” feels very unscientific.

We have changed the relevant phrasing.

Figure 4 – there are only two lines in each panel – is it really necessary to remove error bars for clarity? Or they are very large? This is important information that the readers should be shown.

We agree with the reviewer that including error bars won't reduce the clarity of the figure and have included a ribbon around the line representing error bars. Source data is also being provided with the revised manuscript draft.

The data presented in figure 6 shows the resistance increasing from 0.01% to ~37% in the presence of B. subtilis, this is very different from the data presented in Figure 1

where the frequency of resistance does not change at all. Why is the effect so different between the two figures? Is the effect presented in figure 1 repeatable?

The wildtype strain used in figure 6 is different from the one used in the rest of the figures. For figure 6, we used the parent strain from Bacillus genomic stock center as wildtype since that is more appropriate as the control because the *yydf* and *yorD* knockouts were made in that background. We had included this information in the methods in the manuscript (line 191 in the original draft). However, we understand that this might confuse the readers and hence have included that information more prominently in the legends of figure 6. The effect we see in figure 1 has also been largely replicated in a different experiment in figure 2, so the effect is repeatable.

REVIEWERS' COMMENTS:

Reviewer #2 (Remarks to the Author):

I have reviewed all answers to my comments (as well as those from the other reviewers) and am satisfied with all explanations given by the authors.

Reviewer #3 (Remarks to the Author):

Thank you to the authors for their detailed responses to my comments. I am happy with the changes that the authors have made to the manuscript, and with the justifications provided where no action was taken. These are interesting results and open up the field to further research.